# Survival in Advanced Epithelial Ovarian Cancer Associated with Cardiovascular Comorbidities and Type 2 Diabetes Mellitus

**Stanislav Slavchev** [1,2], **Yavor Kornovski** [1,2], **Angel Yordanov** [3,*], **Yonka Ivanova** [1,2], **Stoyan Kostov** [1,2] and **Svetoslava Slavcheva** [4]

1   Department of Obstetrics and Gynecology, Faculty of Medicine, Medical University "Prof. Dr. Paraskev Stoyanov", 9002 Varna, Bulgaria; st_slavchev@abv.bg (S.S.); ykornovski@abv.bg (Y.K.); yonka.ivanova@abv.bg (Y.I.); drstoqn.kostov@gmail.com (S.K.)
2   Obstetrics and Gynecology Clinic, St. Anna University Hospital, 9002 Varna, Bulgaria
3   Department of Gynecologic Oncology, Medical University Pleven, 5800 Pleven, Bulgaria
4   ES Cardiology, First Department of Internal Diseases, Faculty of Medicine, Medical University "Prof. Dr. Paraskev Stoyanov", 9002 Varna, Bulgaria; hrisivar@abv.bg
*   Correspondence: angel.jordanov@gmail.com

**Abstract:** Background: Ovarian carcinoma (OC) is usually diagnosed at an advanced stage, necessitating a multimodal approach that includes surgery and systemic therapy. The incidence of OC is approximately five times higher in women over 65 years of age. Cardiovascular comorbidities and type 2 diabetes mellitus, both prevalent at this age, can influence therapeutic strategy and have an adverse effect on survival. Objectives: Our study aimed to determine the impact of cardiovascular diseases and diabetes mellitus on survival in advanced ovarian cancer. Materials and methods: From 2004 to 2012, we retrospectively studied 104 patients with advanced epithelial ovarian cancer (FIGO stage II–IV) who underwent surgical treatment at the Gynecology Clinic, St. Anna University Hospital, Varna, Bulgaria. Patients were followed for an average of 90 (52–129) months. We divided the study population into two groups: those with concurrent cardiovascular diseases and type 2 diabetes mellitus (CVD) and those without these comorbidities (No-CVD group). Overall survival (OS), disease-specific survival (DSS), and disease-free survival (DFS) were compared between groups using Kaplan–Meier survival analysis. Cardiovascular comorbidities and diabetes mellitus were evaluated for their prognostic value for survival using multivariate Cox proportional regression analysis adjusted for age, stage of OC, grade and histological type of the tumor, ascites presence, residual tumor size (RT), performance status, and type of hysterectomy. Results: The Kaplan–Meier analysis showed reduced OS and DSS in the CVD group compared to the No-CVD group. The median OS was 24.5 months (95% CI 18.38 months) and 38 months (95% CI 26, not reached), respectively (Log-rank $p = 0.045$). The median DSS was 25.5 months (95% CI 19.39 months) and 48 months (95% CI 28, not reached), respectively (Log-rank $p = 0.033$). The Cox regression multivariate analysis established a lower (by 68%) overall survival rate for the CVD patient group than the No-CVD group, approaching statistical significance (HR 1.68, 95% CI 0.99, 2.86, $p = 0.055$). Cardiovascular diseases and diabetes were associated with a 79% reduction in DSS (HR 1.79, 95% CI 1.02, 3.13, $p = 0.041$) and a twofold increase in the risk of disease progression (HR 2.05, 95% CI 1.25, 3.37, $p = 0.005$). Conclusions: According to our study, cardiovascular comorbidities and diabetes may adversely affect OC survival. Optimal control of cardiovascular diseases, diabetes mellitus, and their risk factors may be beneficial for patients with advanced OC. Further research involving a larger patient population is necessary to establish these comorbidities as independent prognostic factors.

**Keywords:** ovarian carcinoma; cardiovascular comorbidity; diabetes; survival

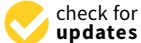



## 1. Introduction

Ovarian cancer is among the ten most common malignancies in women worldwide and the third most common gynecological neoplasm. It is responsible for 4.7% of total cancer mortality [1]. Ovarian cancer ranks eighth as a cause of death among all cancers in women and second among gynecological neoplasms, ranking after cervical cancer [1]. The incidence of ovarian carcinoma (OC) increases with age and is approximately five times higher in women over 65. In developed countries, where life expectancy is longer and populations are aging, OC is more common [2]. Concurrently, the incidence of cardiovascular diseases is higher in older women. Diseases affecting the cardiovascular system are the leading cause of death. They are responsible for 17.9 million deaths per year and 31% of the total mortality for 2016 [3,4]. Cardiovascular morbidity and mortality are prevalent in the diabetic population [5]. Type 2 diabetes itself is among the top 10 causes of death worldwide. Women with diabetes have a 50% higher risk of fatal ischemic heart disease than men [3,5]. Together, cardiovascular and oncological diseases account for more than 60% of total mortality [6].

In recent years, survival in ovarian carcinoma has improved, but this malignancy remains one of the most lethal in women in developed countries [1]. The five-year survival for all stages is 41–48%, and for stages III and IV, in particular, it is even lower—27% and 13.4%, respectively [7,8]. These facts encourage scientists to look for other potentially controllable factors relevant to survival in addition to the well-known prognostic factors such as age, residual tumor size, disease stage, and degree of tumor differentiation [9]. Experts have investigated molecular and genetic changes in different histological types of OC to provide a specific target therapy [10]. In addition, many studies have addressed comorbidity and its correlation with ovarian cancer survival [11–21]. Cardiovascular diseases and diabetes may contribute directly to increased mortality in OC patients and indirectly through treatment selection, suboptimal surgery, and chemotherapy. Our study aimed to evaluate the impact of cardiovascular diseases and diabetes mellitus on survival in patients with advanced epithelial ovarian cancer, given the incidence and prevalence of these comorbidities in the general population. Knowledge of the role of concomitant diseases in the outcome of ovarian carcinoma can guide clinicians in optimizing treatments for comorbid conditions.

## 2. Materials and Methods

### 2.1. Population-Based Study

The presented work is a retrospective study of 104 patients diagnosed with advanced OC (FIGO stages II–IV) and operated at the Obstetrics and Gynecology Clinic, St. Anna University Hospital, Varna, Bulgaria, from 2004 to 2012. We obtained approval and permission to conduct the study from the local ethics committee, NO. 546/30.10.2019.

### 2.2. Pathological Characteristics

The data were collected from medical records and the Bulgarian National Cancer Registry. We assessed the patient population for age, FIGO stage, histological type and grade of OC, residual tumor size, type of hysterectomy (total abdominal hysterectomy, radical hysterectomy and other surgical procedures), type of chemotherapy, ECOG performance status, presence of ascites and lymph node metastases, overall survival (OS) and disease-free survival (DFS). Survival analysis also included the cause of death related to cancer or as a result of another disease, which enabled the determination of specific oncological survival—disease-specific survival (DSS). For the staging of OC, we used the International Federation of Gynecology and Obstetrics (FIGO) classification from 2014. Additionally, we examined the patients for concomitant diseases, particularly for diabetes and cardiovascular diseases (CVD), hypertension, ischemic heart disease (IHD), heart failure, and cerebrovascular disease. The patient population data is presented in Table 1.

**Table 1.** Characteristics of the patient population and comparison of the CVD and No-CVD groups on different variables.

| Characteristic. | Overall, *n* = 104 | No-CVD, *n* = 42 | CVD, *n* = 62 | *p*-Value [1] |
|---|---|---|---|---|
| **Age (years)** | | | | |
| Median (IQR) | 60 (51, 70) | 54 (47, 62) | 65 (58, 72) | **<0.001** |
| Mean (SD) | 60.8 (±11.7) | 55.0 (±10.4) | 64.8 (±10.9) | |
| <55 years | 31 (30%) | 21 (50%) | 10 (16%) | **<0.001** |
| ≥55 years | 73 (70%) | 21 (50%) | 52 (84%) | |
| **Stage, *n* (%)** | | | | 0.5 |
| Stage II | 14 (13%) | 7 (17%) | 7 (11%) | |
| Stage III | 76 (73%) | 28 (67%) | 48 (77%) | |
| Stage IV | 14 (13%) | 7 (17%) | 7 (11%) | |
| **Histology, *n* (%)** | | | | 0.076 |
| Serous | 88 (85%) | 34 (81%) | 54 (87%) | |
| Mucinous | 5 (4.8%) | 1 (2.4%) | 4 (6.5%) | |
| Endometrioid | 1 (1.0%) | 1 (2.4%) | 0 (0%) | |
| Clear cell | 2 (1.9%) | 0 (0%) | 2 (3.2%) | |
| Undifferentiated | 8 (7.7%) | 6 (14%) | 2 (3.2%) | |
| **Grading, *n* (%)** | | | | >0.9 |
| High grade | 100 (96%) | 40 (95%) | 60 (97%) | |
| Low grade | 4 (3.8%) | 2 (4.8%) | 2 (3.2%) | |
| **Residual tumor size, *n* (%)** | | | | 0.9 |
| <1 cm | 44 (42%) | 18 (43%) | 26 (42%) | |
| 1–2 cm | 20 (19%) | 7 (17%) | 13 (21%) | |
| >2 cm | 40 (38%) | 17 (40%) | 23 (37%) | |
| **Performance status, *n* (%)** | | | | 0.7 |
| PS = 0 | 17 (16%) | 9 (21%) | 8 (13%) | |
| PS = 1 | 28 (27%) | 11 (26%) | 17 (27%) | |
| PS = 2 | 26 (25%) | 10 (24%) | 16 (26%) | |
| PS = 3 | 33 (32%) | 12 (29%) | 21 (34%) | |
| PS > 3 | 0 (0%) | 0 (0%) | 0 (0%) | |
| **Ascites, *n* (%)** | | | | 0.062 |
| No ascites | 35 (34%) | 10 (24%) | 25 (42%) | |
| Ascites present | 67 (66%) | 32 (76%) | 35 (58%) | |
| Unknown | 2 | 0 | 2 | |
| **Lymph node metastasis, *n* (%)** | | | | >0.9 |
| No LN metastasis | 25 (24%) | 11 (26%) | 14 (23%) | |
| LN metastasis | 33 (32%) | 13 (31%) | 20 (32%) | |
| LN not evaluated | 46 (44%) | 18 (43%) | 28 (45%) | |
| **Type of surgery, *n* (%)** | | | | 0.058 |
| TH + BSO | 23 (22%) | 11 (26%) | 12 (19.3%) | |
| Radical hysterectomy | 74 (71%) | 31 (74%) | 43 (69.4%) | |
| Other | 7 (7%) | 0 (0%) | 7 (11.3%) | |
| **Chemotherapy, *n* (%)** | | | | 0.10 |
| Adjuvant | 97 (93%) | 40 (95.2%) | 57 (92%) | |
| Neoadjuvant | 3 (2.9%) | 0 (0%) | 3 (4.8%) | |
| Palliative | 2 (1.9%) | 2 (4.8%) | 0 (0%) | |
| No chemotherapy | 2 (1.9%) | 0 (0%) | 2 (3.2%) | |
| **Overall survival** (months), Median (95% CI) | 30 (23, 40) | 38 (26, not reached) | 24.5 (18, 38) | **0.045** |
| **5-year OS probability**, (%) (95%CI)) | 28 (20, 38) | 38 (25, 56) | 21 (13, 34) | |
| **DFS** (months), Median (95% CI) | 14 (9, 26) | 28 (14, 36) | 10.5 (9, 21) | **0.006** |
| **3-year DFS probability**, (%) (95%CI)) | 21 (15, 31) | 31 (20, 49) | 15 (8, 27) | |
| **DSS** (months), Median (95% CI) | 34 (24, 45) | 48 (28, not reached) | 25.5 (19, 39) | **0.033** |
| **5-year DSS probability**, (%) (95%CI)) | 29 (21, 40) | 40 (28, 60) | 22 (13, 35) | |

[1] Wilcoxon rank-sum test; Pearson's Chi-squared test; Fisher's exact test. Values in bold represent significant *p* values. CVD—Cardiovascular diseases and diabetes; IQR—Interquartile range; SD—Standard deviation; PS—Performance status; LN—Lymph nodes; TH—Total hysterectomy; BSO—Bilateral salpingo-oophorectomy; OS—Overall survival; DFS—Disease-free survival; DSS—Disease-specific survival.

The patient population was divided into two groups: one group without cardiovascular diseases and diabetes (No-CVD) and another group with these diseases (CVD). The

cardiovascular diseases (CVD) group also included patients with diabetes mellitus only, invariably associated with cardiovascular damage. The two groups were compared in terms of ovarian carcinoma stage, grade and histological type, residual tumor size, ascites, lymph node metastases, performance status, hysterectomy type, and systemic therapy.

### 2.3. Statistical Analysis

We performed statistical data processing using the program R, version 4.0.3 (10 October 2020). Descriptive analysis was applied; continuous variables were presented as the mean and standard deviation (SD), median and interquartile range (IQR) 25–75%, and categorical variables as relative frequencies with percentages. The Wilcoxon rank-sum test for continuous variables, Pearson's Chi-squared test for categorical variables, and Fisher's exact test were used to compare the CVD and No-CVD groups. Survival analysis was performed using Kaplan–Meier statistics with Log Rank, Breslow, and Tarone-Ware tests to compare survival probability. We carried out Cox univariate and multivariate regression analyses to assess the predictability of the various factors for survival. In addition, we also used univariate and multivariate Poisson regression statistics to identify the factors associated with surgical treatment success and optimal residual tumor size. A two-sided probability value under 0.05 was considered statistically significant.

### 3. Results

The mean age of the study population was 60.8 (SD $\pm$ 11.7) years. Most ovarian carcinoma patients were in FIGO stage III (73%) with a predominance of serous histological type (83%) and high-grade tumors (96%). There were no detected moderately differentiated (grading 2) tumors due to the small percentage of endometrioid histological variants. Seventy-one percent of patients were operated on through radical hysterectomy with retroperitoneal approach, bilateral adnexectomy, and omentectomy. Twenty-two percent underwent total abdominal hysterectomy with bisalpingo-oophorectomy and omentectomy (THO + BSO), and the rest (7%) were subjected to a different type of surgery—diagnostical or adnexectomy—only in cases of previous hysterectomy. The goal of cytoreductive surgery in our population was to achieve a residual tumor size (RT) of less than 1 cm. Optimal cytoreduction (RT < 1 cm) was achieved in 42% of the patients. Lymph node dissection—pelvic and/or paraaortic—was performed on 58 (56%) patients, and approximately half of them had lymph node involvement (32% of the total population). The lymph nodes were not histologically verified in 46% of the patients. Therefore, we did not include lymph node involvement in the multivariate regression analysis of survival. Ascites were present in approximately 2/3 of the patients. Adjuvant 6-cycle combination chemotherapy was administered to 93% of the patients who received carboplatin at a dose of AUC 5–7.5, and paclitaxel 175 mg/m$^2$ was administered every three weeks, or with an alternative regimen—a combination of paclitaxel 135 mg/m$^2$ day 1 and 60 mg/m$^2$ day 8 with cisplatin 75–100 mg/m$^2$ every three weeks. Neoadjuvant chemotherapy with paclitaxel 175 mg/m$^2$ and carboplatin AUC 6–5 was performed on a small number of patients. The median overall survival (OS) of the patients was 30 months (IQR 23, 40), the 5-year overall survival was 28% (95% CI 20, 38%). The median disease-specific survival was 34 months (IQR 24, 45 months), the 5-year DSS was 29% (95% CI 21, 40%). The median disease-free survival (DFS) was 14 months (IQR 9, 26 months), the 3-year DFS was 21% (95% CI 15, 31%). We identified 91 patients with OC progression during a mean follow-up of 90 months (52–129 months), 78 deaths overall, and 73 deaths due to the malignancy. The summarized data is presented in Table 1.

We also evaluated the patient population for concomitant diseases, primarily for the presence of cardiovascular diseases and diabetes type 2 (Table 2). Sixty-two (59%) patients had various cardiovascular diseases with or without diabetes mellitus and were classified as a CVD group. Forty-two (41%) patients did not have any of the diseases mentioned above and became part of the No-CVD group. Other concomitant diseases such as thyroid disorders, gastrointestinal tract diseases (ulcer disease, cholelithiasis, chronic

hepatitis), lung diseases (COPD, bronchial asthma, pulmonary fibrosis), kidney diseases (nephrolithiasis), chronic venous insufficiency, epilepsy were also present among patients. These comorbidities were present in both the CVD and No-CVD groups but with a tiny relative proportion and were not included in the survival analysis.

**Table 2.** Distribution of cardiovascular diseases and diabetes mellitus among patients.

| Concomitant Diseases | *Overall, n* = **104** [1] |
|---|---|
| Diabetes only | 3 (2.9%) |
| Hypertension | 30 (28%) |
| Ischemic heart disease | 12 (12%) |
| Heart failure, cerebral vascular disease | 7 (6.8%) |
| CVD + diabetes | 10 (9.7%) |
| No CVD | 42 (41%) |
| Diabetes overall proportion | 13 (12.5%) |

[1] The distribution of diseases is denoted by *n* (%).

The comparative analysis between the two groups showed no statistically significant difference between them in terms of the different variables (Table 1), except for age. The CVD group's median age was significantly higher than the No-CVD group, with a difference of 11 years on average (95% CI 5.5, 14.0 $p < 0.001$). The reason behind this variance is that cardiovascular diseases are common in older women over 50–55 years of age. This fact requires not only univariate but also multivariate Cox regression analysis, which takes into account the influence of cardiovascular comorbidities, as well as other factors such as age, stage, grade of the tumor, the presence of ascites, residual tumor size, type of surgical procedure, and performance status (PS).

The Kaplan–Meier analysis (Figure 1) showed a statistically significant difference in overall survival between CVD and No-CVD groups at a median of 24.5 months (95% CI 18, 38 months) and 38 months (95% CI 26, not reached) (Log-rank $p = 0.045$), respectively. Regarding cancer-specific survival (DSS), the median survival of the CVD group was 25.5 months (95% CI 19, 39 months), and the non-CVD group was 48 months (95% CI 28, not reached), with a statistically significant difference between the groups (Log-rank $p = 0.033$). The comparison of the groups with the Breslow and Tarone-Ware methods did not show a statistically significant difference, which means that a difference in survival was not observed in the initial months after diagnosing ovarian cancer and at the beginning of treatment. There was a significant difference (Log-rank $p = 0.006$; Breslow $p = 0.0036$; Tarone-Ware $p = 0.0016$) in terms of disease-free survival (DFS) for the CVD group, the median DFS was 10.5 months (95% CI 9, 21 months), and for the No-CVD group—28 months (95% CI 14, 36 months) (Figure 2).

According to the univariate Cox regression analysis, the following factors were unfavorable for overall survival: an advanced stage of the disease (stage IV), residual tumor size over 2 cm, performance status (PS = 3), and the presence of cardiovascular diseases with or without diabetes. Age had no statistical significance as a predictor of overall survival (Table 3). We analyzed these parameters with multivariate Cox regression analysis and established a lower (by 68%) overall survival probability in patients with CVD compared to patients without CVD, approaching statistical significance (HR 1.68, 95% CI 0.99, 2.86, $p = 0.055$) (Table 3). Regarding disease-specific survival, a large residual tumor size (RT > 2 cm) and a high-performance status (PS = 3) had the highest predictive values and were associated with a significant reduction in survival: for PS = 3—HR 2.23, 95% CI 1.01, 4.95, $p = 0.048$, and for OT > 2 cm—HR 2.34, 95% CI 1.27, 4.30, $p = 0.006$ (Figure 3). Cardiovascular comorbidities and diabetes were associated with a 79% reduction in specific survival (DSS) when statistical significance was reached (HR 1.79, 95% CI 1.02, 3.13, $p = 0.041$) (Figure 3). Cardiovascular diseases and diabetes mellitus were also unfavorable predictors of disease-free survival by increasing the risk of disease progression by twofold (HR 2.05; 95% CI 1.25, 3.37; $p = 0.005$) (Figure 4). Univariate and multivariate Poisson regression analysis showed that cardiovascular diseases did not affect the outcome of

surgical treatment measured by the residual tumor size (univariate analysis—IRR 0.99, CI 0.75, 1.31, *p* = 0.93; multivariate analysis—IRR 0.98, CI 0.72, 1.35, *p* = 0.9) (Figure 5).

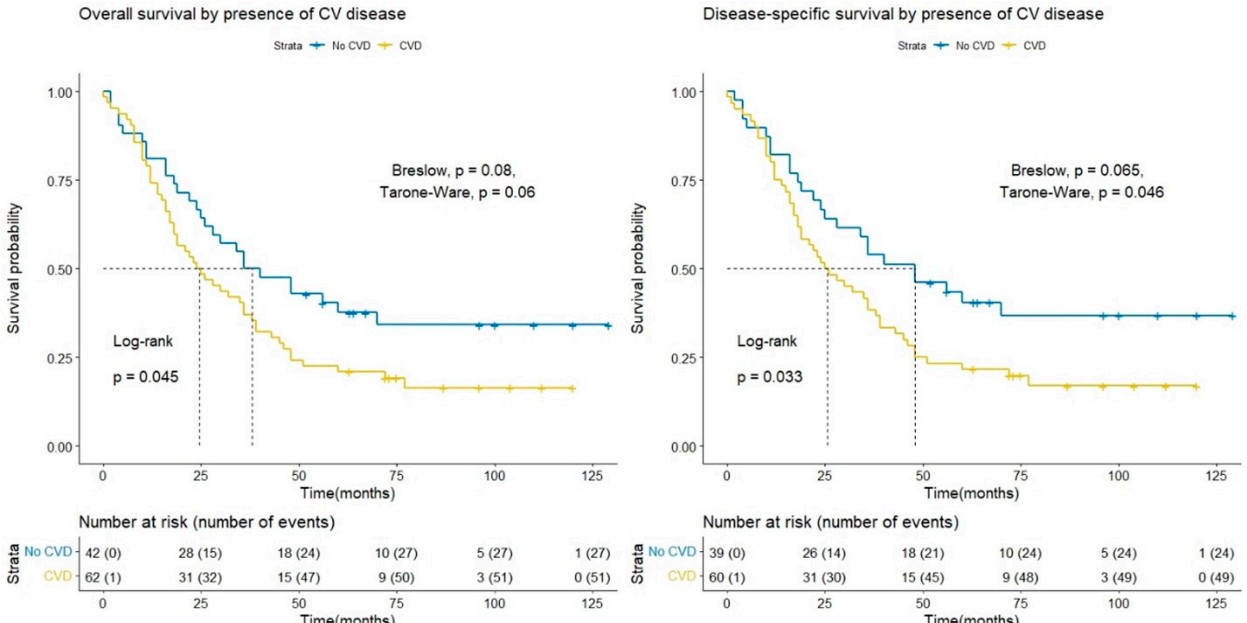

**Figure 1.** Comparison of CVD and Non-CVD groups in terms of overall and specific survival by Kaplan–Meier survival analysis.

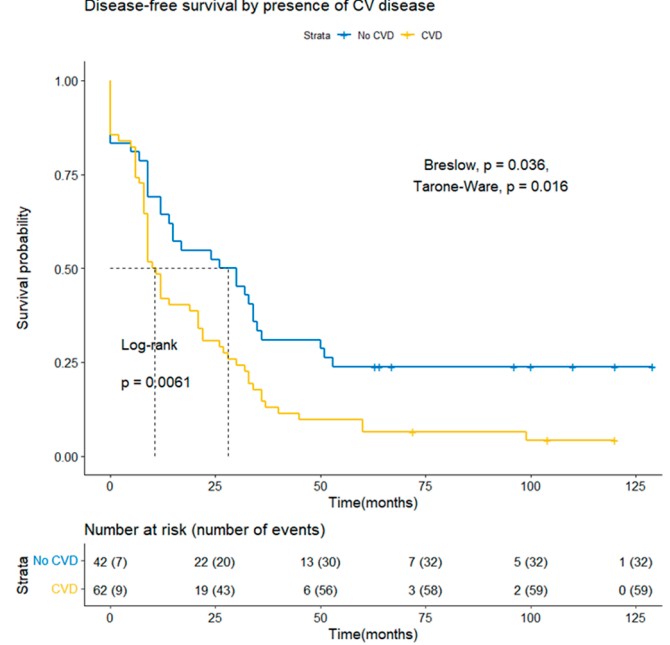

**Figure 2.** Comparison of CVD and No-CVD groups for disease-free survival by Kaplan–Meier survival analysis.

**Table 3.** Univariate and multivariate Cox regression analysis of overall survival.

| Characteristic | Univariate Analysis | | | Multivariate Analysis | | |
|---|---|---|---|---|---|---|
| | HR [1] | 95% CI [1] | *p*-Value | HR [1] | 95% CI [1] | *p*-Value |
| **Age (years)** | 1.01 | 0.99, 1.03 | 0.6 | 0.99 | 0.97, 1.01 | 0.5 |
| **Stages** | | | | | | |
| Stage II | — | — | | — | — | |
| Stage III | 2.08 | 0.94, 4.57 | 0.069 | 1.51 | 0.61, 3.75 | 0.4 |
| Stage IV | 5.14 | 2.01, 13.1 | **<0.001** | 3.35 | 1.06, 10.6 | **0.039** |
| **Ascites** | | | | | | |
| No ascites | — | — | | — | — | |
| With ascites | 1.25 | 0.77, 2.01 | 0.4 | 1.03 | 0.61, 1.73 | >0.9 |
| **Type of surgery** | | | | | | |
| TH + BSO | — | — | | — | — | |
| Radical hysterectomy | 1.05 | 0.60, 1.83 | 0.9 | 0.99 | 0.51, 1.93 | >0.9 |
| Other | 1.79 | 0.74, 4.38 | 0.2 | 0.87 | 0.30, 2.56 | 0.8 |
| **Residual tumor size** | | | | | | |
| RT < 1 cm | — | — | | — | — | |
| RT 1–2 cm | 1.77 | 0.95, 3.32 | 0.073 | 1.76 | 0.90, 3.45 | 0.10 |
| RT > 2 cm | 2.82 | 1.68, 4.73 | **<0.001** | 2.17 | 1.21, 3.90 | **0.009** |
| **Grading** | | | | | | |
| Low grade | — | — | | — | — | |
| High grade | 4.91 | 0.68, 35.4 | 0.11 | 2.21 | 0.29, 17.0 | 0.4 |
| **LN metastasis (*n* = 58)** | | | | | | |
| No LN metastasis | — | — | | | | |
| With LN metastasis | 1.47 | 0.77, 2.80 | 0.2 | | | |
| **Performance status** | | | | | | |
| 0 | — | — | | — | — | |
| 1 | 1.12 | 0.53, 2.36 | 0.8 | 1.34 | 0.61, 2.95 | 0.5 |
| 2 | 1.48 | 0.70, 3.12 | 0.3 | 1.87 | 0.84, 4.16 | 0.13 |
| 3 | 2.76 | 1.37, 5.56 | **0.005** | 2.40 | 1.09, 5.27 | **0.029** |
| **CVD** | | | | | | |
| No CVD | — | — | | — | — | |
| With CVD | 1.60 | 1.00, 2.56 | **0.049** | 1.68 | 0.99, 2.86 | 0.055 |

[1] HR = Hazard Ratio; CI = Confidence Interval. Statistical power = 0.164. Significant *p*-values are in bold.

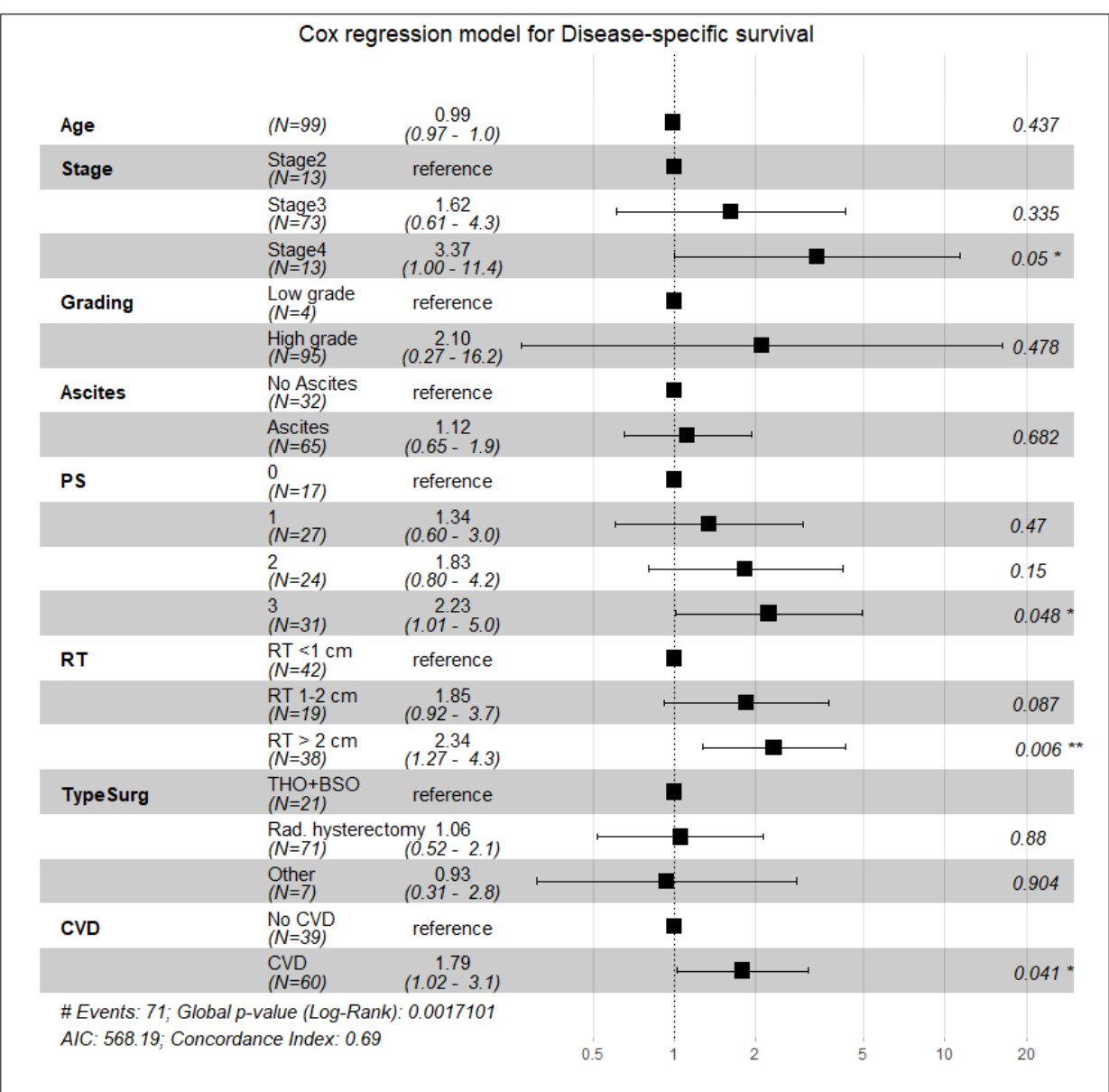

**Figure 3.** Forest plot. Cox regression multivariate analysis of the influence of various variables and cardiovascular diseases on disease-specific survival. Statistical power = 0.19. ( * $p \leq 0.05$, ** $p \leq 0.01$; # Events = number of events).

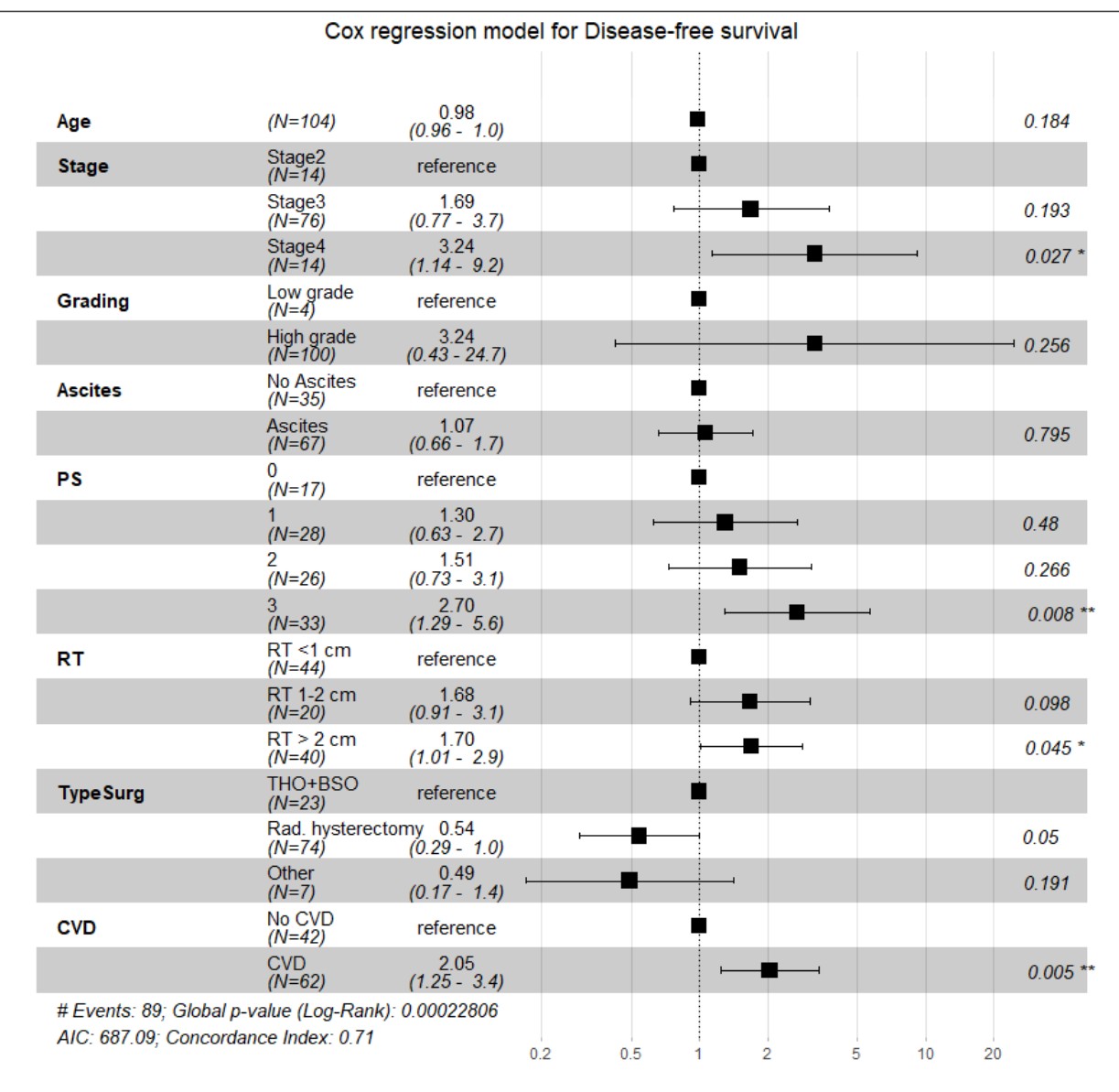

**Figure 4.** Forest plot. Cox regression multivariate analysis of the impact of various variables and cardiovascular diseases on disease-free survival. Statistical power 0.27. (* $p \leq 0.05$, ** $p \leq 0.01$; # Events = number of events).

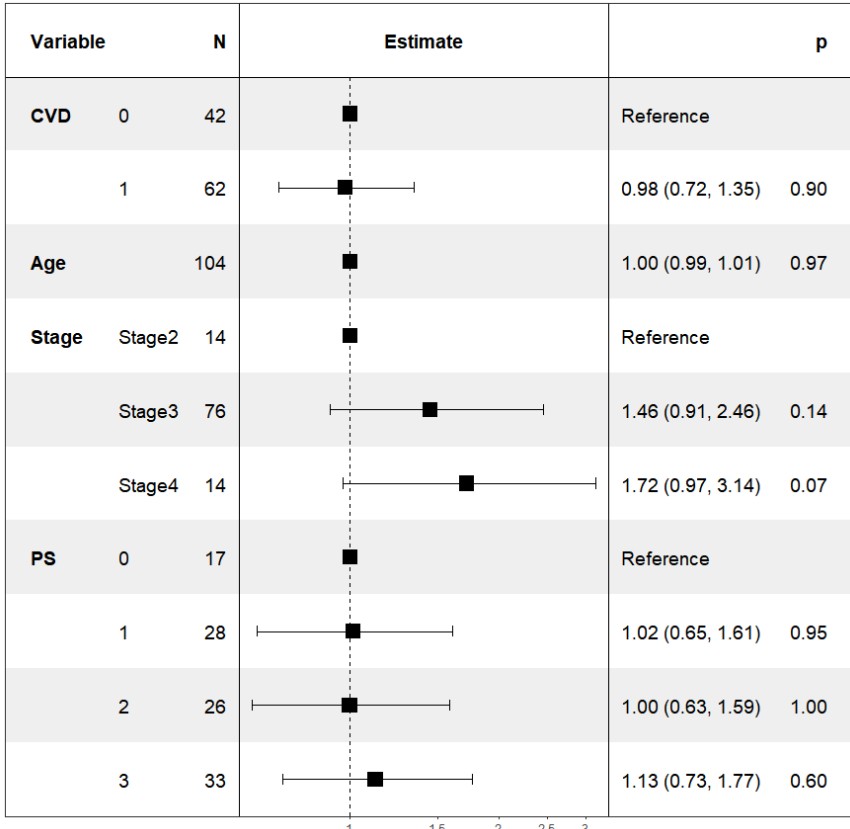

**Figure 5.** Poisson multivariate regression analysis assessing the impact of various factors on the success of surgical treatment, measured by the residual tumor size.

## 4. Discussion

Cardiovascular diseases and diabetes mellitus are common among patients with neoplastic diseases, and their frequency is between 12 and 60% [11,22]. The relationship between cancer and cardiovascular diseases is multidirectional. A correlation is found at both the pathophysiological and pathogenetic level. Common molecular pathways, genetic mutations, metabolic disorders, and lifestyle risk factors are involved in the onset and development of these two types of diseases [23]. The incidence of ovarian epithelial carcinoma increases with age; the average age at cancer diagnosis is 60–65 years [24]. Older women are more likely to have elevated BMI, insulin resistance, and hyperinsulinemia, which are risk factors for cardiovascular disease and ovarian cancer by activating systemic inflammation and oxidative stress [25]. Diabetes, hyperinsulinemia, and insulin resistance affect the biological processes in cancer cells through increased levels of insulin-like growth factor and activation of molecular pathways responsible for cell proliferation, thus contributing to tumor progression [17,25,26]. Atherosclerosis, the dominant underlying cause of various cardiovascular diseases, represents chronic vascular inflammation involving various intracellular signaling mechanisms. These molecular pathways are also involved in the onset and progression of various cancer types, including ovarian cancer [23,27].

At the same time, some medications used to prevent and treat cardiovascular diseases and diabetes also have a preventive effect on developing cancer diseases. Such a potential role has been attributed to statins, aspirin, angiotensin-converting enzyme inhibitors, and metformin, which have been shown to affect angiogenesis and tumor cell proliferation [23,28]. In addition, metformin has been shown to inhibit the growth of ovarian cancer cell lines, potentiate the effect of cisplatin, reduce the risk of ovarian carcinoma, and increase survival in OC patients with diabetes [29]. Beyond the preventive effect of statins on the development of ovarian cancer, some authors report their beneficial effect on survival [30].

The likelihood of delayed cancer diagnosis is an additional aspect of concomitant diseases, particularly cardiovascular diseases. Tetsche et al. [12] found that comorbidity is associated with a more advanced stage of the disease. This fact could be explained by the biological behavior of the tumor in the conditions of another chronic disease or by late diagnosis in the presence of concomitant diseases [12]. Cardiovascular comorbidities could influence decisions regarding therapeutic strategy or compromise cytoreductive surgery and systemic therapy. According to a study by Janssen-Heijnen et al. [11], the recommended surgical and chemotherapeutic treatments were implemented in a significantly lower percentage of patients with FIGO stage II–III OC who had at least one concomitant disease, regardless of age, over or under 70 years old [11]. Mallen et al. [13] compared older OC patients over 70 with those younger than 70. The authors believed that concomitant diseases were not responsible for a worse prognosis in adults. According to the authors, the different tumor biology, characterized by greater aggressiveness and resistance to platinum, contributed to lower survival of the adult population, which also had more severe comorbidities (including cardiovascular diseases and diabetes mellitus) [13]. The researchers found that the lower the probability of optimal cytoreduction, the higher the chance of an inability to complete adjuvant chemotherapy, and more frequent resistance to platinum therapy were independent adverse factors in adults [13].

Standard systemic chemotherapy for OC includes a combination of carboplatin and paclitaxel, and since recently, PARP (poly ADP-ribose polymerase) inhibitors (Olaparib, niraparib) and anti-angiogenic agents (bevacizumab) have been introduced as first-line therapy [31]. Systemic therapy medications are associated with a potential risk of cardio-vascular complications, referred to as cardiotoxicity. The risk of developing cardiotoxicity increases in patients who have pre-existing cardiovascular risk factors and diseases [32]. Various cardiac and vascular complications, such as heart failure, arrhythmias, myocardial ischemia, and pericardial diseases, have been reported with the use of taxanes and platinum drugs. Bevacizumab has been related to worsening hypertension and the development of arterial thrombosis [33–35]. Simultaneously, experimental data suggest that Olaparib may have a cardioprotective effect [36].

Most researchers who have studied the impact of comorbidities on ovarian cancer survival consider cardiovascular diseases in conjunction with other comorbidities incorporated into the Charlson Comorbidity Index (CCI) or Ovarian Cancer Comorbidity Index (OCCI), assessing primarily overall survival. A meta-analysis of eight prospective studies involving more than 12,000 patients demonstrated an increased mortality risk in patients with concomitant diseases (HR 1.2, 95% CI 1.11–1.3) [13]. Noer et al. (2018) confirmed that comorbidity was significantly associated with worse overall survival HR 1.36 (1.05–1.77), $p = 0.020$ [16].

Few studies have explicitly focused on cardiovascular diseases and diabetes as potential factors in ovarian cancer survival. Not all have reported the definite prognostic significance of cardiovascular comorbidities for reduced survival, assessing overall rather than disease-specific survival. Minlikeeva et al. [17] studied the effects of hypertension, heart disease, and diabetes on overall survival in OC patients, using data from 15 studies involving over 9000 patients. The authors found no evidence of an increased mortality risk in patients with hypertension or heart disease (for hypertension HR = 0.95; 95% CI = 0.88–1.02; for CVD 0.99, 95% CI = 0.75–1.30, respectively). The presence of diabetes mellitus was associated with reduced ovarian cancer survival (HR = 1.12; 95% CI = 1.01–1.25). The authors identified diabetes mellitus as an independent predictor relevant to ovarian cancer prognosis [17]. Akhavan et al. [18] confirmed the above by examining the effects of diabetes on OC survival. After adjusting for other major prognostic factors, including the use of metformin, the authors reported that diabetes was significantly associated with a worse prognosis for overall survival (HR 3.93, 95% CI 2.01 to 7.68; $p < 0.001$) and disease-free survival (HR 3.93, 95% CI 2.01 to 7.68; $p < 0.001$). Interestingly, the authors observed a lower degree of tumor differentiation and a more advanced stage in patients with diabetes mellitus [18]. Similar results were recorded by Ferriss et al. [19], according to which dia-

betes is an independent unfavorable predictor and leads to a 1.8-times higher mortality risk in patients with advanced OC, despite the achieved optimal cytoreduction (HR 1.8, 95% CI 1.02, 3.1, *p* = 0.042). The researchers did not find evidence associating hypertension with OC survival [19]. However, they reported that patients referred for neoadjuvant chemotherapy treatment and interval debulking surgery had a higher incidence of hypertension and diabetes [19]. Patients who had never undergone surgical treatment had a higher incidence of concomitant diseases—diabetes, hypertension, heart and lung diseases, and venous thromboembolism [19].

Other researchers have analyzed survival factors in patients undergoing radical pelvic surgery (*n* = 646) for a variety of malignant tumors, including ovarian carcinoma (*n* = 57) [20]. According to them, a high ASA score, cardiovascular comorbidities, treatment for recurrent cancer, ovarian cancer, pulmonary thromboembolism, and acute respiratory distress syndrome were all associated with poor 5-year overall survival [20]. Shinn et al. [21] examined emerging cardiovascular events (venous thromboembolism, pulmonary hypertension, ischemic heart disease, cerebrovascular disease, pericardial effusion and tamponade, valvular heart disease, cardiomyopathy, and heart failure) during treatment and follow-up for OC. The authors concluded that only venous thromboembolism and pulmonary hypertension were independent predictors of reduced survival after adjusting for stage, grading, and level of cytoreduction [21]. The studies discussed previously, particularly those involving concomitant cardiovascular pathologies and diabetes mellitus, have different designs and populations. The study with the largest examined population did not present results of multivariate analysis adjusted for residual tumor size.

Our study focused on some comorbidities, namely cardiovascular diseases and diabetes, considering their high prevalence, social significance, and pathophysiological relationship with oncological diseases. Cardiovascular pathology and diabetes mellitus, present at the time of ovarian carcinoma detection, are associated with poorer overall survival (HR 1.68, 95% CI 0.99, 2.86, *p* = 0.055) and disease-specific survival (HR 1.79, 95% CI 1.02, 3.13, *p* = 0.041), respectively, with no and borderline statistical significance. Concurrently, these comorbidities double the risk of disease progression (HR 2.05, 95 % CI 1.25, 3.37, *p* = 0.005). According to the Cox regression models represented, CVDs and diabetes are significant predictors of poor OC survival (DSS (*p* = 0.041) and DFS (*p* = 0.005)). On the other hand, due to the small sample size, the statistical power of these Cox multivariate regression analyses is low. With the study's limited statistical power, we cannot assert that cardiovascular diseases and diabetes are independent predictors of increased OC mortality in the same way that other well-established factors such as residual tumor size, FIGO stage, and ECOG PS are. We propose that future research on larger patient populations should evaluate the influence of cardiovascular comorbidities and diabetes on OC outcomes in isolation from other comorbidities.

All of our patients underwent surgical treatment and were referred for surgery following a preliminary evaluation. Due to these constraints, we were unable to assess the impact of comorbidities on treatment choice. We used the residual tumor size as a surrogate marker for the success of surgical treatment. The presence of cardiovascular disease and diabetes did not prove to be factors affecting the results of surgery and the degree of cytoreduction (IRR 0.98, CI 0.72, 1.35, *p* = 0.9).

CVDs and diabetes both have a negative effect on OC survival in our patient cohort, especially on DFS and DSS, which may be explained by pathophysiologic processes that contribute to tumor progression by creating a chronic inflammatory environment. Further, the coexistence of these diseases may have resulted in the delay or under-dosage of individual chemotherapy courses, ultimately leading to OC relapse. Considering these speculations, optimal control of diabetes, cardiovascular diseases, and risk factors may support ovarian cancer treatment. Additionally, certain medications used to treat these comorbidities, including statins, aspirin, metformin, and angiotensin-converting enzyme inhibitors, may have anticancer properties through their effects on chronic inflammation and cell proliferation [23,28,29].

Our single-center study is limited by its retrospective design and the small number of patients with advanced ovarian carcinoma—a less common type of cancer than cervical and endometrial cancers. This resulted in the Cox multivariate regression analyses obtaining a low statistical power. Because the selected population included patients who had undergone surgery, no conclusions regarding the entire population can be drawn. Cytoreductive surgery may have been avoided in a subset of the general population, owing to severe concurrent pathology. An advantage of our study is the long follow-up period, which averaged 90 months, with a maximum period of 129 months. It is important to note that we assessed specific cancer survival (DSS), which has been investigated in terms of comorbidity by very few researchers. In addition, the multivariate model took into account the effect of cardiovascular diseases on survival after adjusting for the influence of other established risk factors—age, stage, grading, residual tumor size, and performance status.

## 5. Conclusions

Ovarian carcinoma is usually detected late, at an advanced stage, when a combination of surgical and systemic long-term treatment is required. In this regard, therapy success may be contingent on comorbidity. This is especially relevant in the presence of cardiovascular diseases and diabetes mellitus, which are common in the elderly population. They may be limiting factors for optimal therapy. At the same time, evidence suggests that they are associated with more aggressive tumor biology. According to our study, cardiovascular comorbidities and diabetes may contribute to poor OC survival. The strict control of diabetes, coexisting cardiovascular disease, and risk factors is essential for cancer treatment success. Further research involving a larger patient population is necessary to establish these comorbidities as independent prognostic factors.

**Author Contributions:** Conceptualization, S.S. (Stanislav Slavchev) and S.S. (Svetoslava Slavcheva); methodology, S.K. and Y.I.; formal analysis, S.S. (Stanislav Slavchev), Y.I. and S.K.; investigation, S.K. and Y.I.; resources, Y.I and S.K.; data curation, S.S. (Stanislav Slavchev); writing—original and draft preparation, S.K. and S.S. (Svetoslava Slavcheva); writing—review and editing, S.S. (Stanislav Slavchev), S.S. (Svetoslava Slavcheva), A.Y. and Y.I.; visualization, A.Y; supervision A.Y., Y.K. All authors have read and agreed to the published version of the manuscript.

**Funding:** This research received no external funding.

**Institutional Review Board Statement:** The study was conducted according to the guidelines of the Declaration of Helsinki, and approved by the Institutional Ethics Committee of Medical University Varna (NO. 546/30 October 2019).

**Informed Consent Statement:** Informed consent was obtained from all subjects involved in the study.

**Data Availability Statement:** The data presented in this study are available on request from the corresponding author.

**Conflicts of Interest:** The authors declare no competing interests.

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
