# Peer review of "Survival in Advanced Epithelial Ovarian Cancer Associated with Cardiovascular Comorbidities and Type 2 Diabetes Mellitus"

_curroncol, doi:10.3390/curroncol28050313_

Round 1
Reviewer 1 Report
Accept as it stands after English language editing
Author Response
"Please see the attachment"

Reviewer 2 Report
The topic of the study is very important, but I have some doubts regarding methodological aspects of the study.
1). I am not convinced that it was right to include patients with diabetes alone in the population of patients with CVD, there were only 3 patients with alone diabetes, but in such a small population they could influenced the result. Some results are borderline in terms of statistical significance. I would recommend that patients with diabetes alone should be rejected from the analyzes.
2). The authors mention that they had patients with chronic kidney disease. Why have they been not included in the CVD population on a similar basis to diabetes? How many patients have you have had with kidney diseases?
3). The authors should provide more details on how many cancer disease progressions were recorded during follow up, whether there were consecutive lines of anticancer treatment (chemotherapy or targeted therapy) after progression, and how many deaths there were observed. Everything is worth showing when comparing patients with CVD vs nonCVD. Such a presentation would enable better planning of multivariate analyzes.
4). The authors should add information about the cardiological and anti-diabetic drugs taken. In the discussion, the authors themselves write that these drugs may have prognostic value in oncology.
Author Response
"Please see the attachment"

Reviewer 3 Report
In my opinion, this study is well conducted and described. The authors aim to compare suvival between CVD and non-CVD patients. I would suggest a moderate revision for English style and some errors and typos. However, the main concern about this study is the low sample size. The authors discussed on this topic in the discussion section, but I would suggest to provide the calculation of the statistical power, especially for the Cox regression.
Author Response
"Please see the attachment"

Round 2
Reviewer 2 Report
Authors explained all limitations of the study.
Author Response
Dear reviewer,
Thank you for your review.
Reviewer 3 Report
In my opinion, The statistical power is too low to be published
Author Response
Dear reviewer
Thank you for your review